# Laws Restricting Access to Abortion Services and Infant Mortality Risk in the United States

**DOI:** 10.3390/ijerph17113773

**Published:** 2020-05-26

**Authors:** Roman Pabayo, Amy Ehntholt, Daniel M. Cook, Megan Reynolds, Peter Muennig, Sze Y. Liu

**Affiliations:** 1School of Public Health, University of Alberta, 11405-87, Edmonton, AB T6G 1C9, Canada; 2School of Community Health Sciences, University of Nevada, Reno, NV 89557, USA; ehntholt@gmail.com (A.E.); dmcook@unr.edu (D.M.C.); 3Sociology Department, University of Utah, Salt Lake City, UT 84117, USA; megan.reynolds@soc.utah.edu; 4Mailman School of Public Health, Columbia University, New York, NY 10032, USA; m124@cumc.columbia.edu; 5Public Health Department, Montclair State University, Montclair, NJ 07043, USA; lius@montclair.edu

**Keywords:** US state laws, abortion, infant mortality

## Abstract

*Objectives:* Since the US Supreme Court′s 1973 *Roe v. Wade* decision legalizing abortion, states have enacted laws restricting access to abortion services. Previous studies suggest that restricting access to abortion is a risk factor for adverse maternal and infant health. The objective of this investigation is to study the relationship between the type and the number of state-level restrictive abortion laws and infant mortality risk. *Methods:* We used data on 11,972,629 infants and mothers from the US Cohort Linked Birth/Infant Death Data Files 2008–2010. State-level abortion laws included Medicaid funding restrictions, mandatory parental involvement, mandatory counseling, mandatory waiting period, and two-visit laws. Multilevel logistic regression was used to determine whether type or number of state-level restrictive abortion laws during year of birth were associated with odds of infant mortality. *Results:* Compared to infants living in states with no restrictive laws, infants living in states with one or two restrictive laws (adjusted odds ratio (AOR) = 1.08; 95% confidence interval [CI] = 0.99–1.18) and those living in states with 3 to 5 restrictive laws (AOR = 1.10; 95% CI = 1.01–1.20) were more likely to die. Separate analyses examining the relationship between parental involvement laws and infant mortality risk, stratified by maternal age, indicated that significant associations were observed among mothers aged ≤19 years (AOR = 1.09, 95% CI = 1.00–1.19), and 20 to 25 years (AOR = 1.10, 95% CI = 1.03–1.17). No significant association was observed among infants born to older mothers. *Conclusion:* Restricting access to abortion services may increase the risk for infant mortality.

## 1. Introduction

The infant mortality rate (IMR), the number per 1000 live births of infant deaths before the age of 1, is one of the best predictors of a nation′s life expectancy and widely used as an indicator of population health [1]. In 2011, around 24,000 infants died in the United States (US), resulting in an IMR of 6.1, nearly twice the Organization for Economic Co-operation and Development (OECD) average of 3.4/1000 [2]. Known risk factors for infant mortality include individual-level risk factors (e.g., socio-economic status, such as mother’s education or household income) and larger contextual factors (e.g., neighborhood environment or state-level policies that affect access to health care) [3].

Within the US, IMRs vary greatly across socio-demographic groups. Infants born to Black mothers, single mothers, and low-income mothers have higher IMRs compared to infants born to White mothers, married mothers, and moderate or high-income mothers [4]. For example, in 2016, the IMR among non-Hispanic Blacks was 11.4 per 1000, whereas the IMR for non-Hispanic Whites was 4.9 [5]. These socio-demographic groups may have limited access to resources, such as psychosocial supports, material resources, and vital health and reproductive services.

Abortion, a medical procedure terminating a pregnancy, is recognized as a key component of reproductive health services [6], which the United Nations recognizes as an important predictor of national well-being and population health, generally, and maternal and infant mortality, specifically [7,8,9]. Beyond the abortion laws examined in this study, states continue to create policy restricting access to safe abortions. Currently, the Supreme Court is hearing one such case affecting a provider’s rights to provide abortion services. Beyond the immediate constitutional value of any restrictive abortion law, courts should consider the health impact of such laws on women and their future unborn children.

Restrictive abortion policies may have a detrimental effect on both maternal and infant health via several mechanisms. First, restrictive abortion policies may jeopardize patient health by undermining providers′ medical counsel. For example, some US states require counseling that provides inaccurate information about negative mental health consequences of abortion or a link between abortion and increased risk of breast cancer [10]. Second, restrictions may increase psychological distress, which is a known risk factor for poor birth outcomes. Both human and animal studies show that psychological stress produces a cascade of neuroendocrine changes that increase the risk of serious birth complications (e.g., pre-eclampsia) and thereby increase infant mortality [11,12,13,14].

Third, these policies may increase the risk for postpartum depression by inducing dramatic, but unwanted life changes. Postpartum depression can contribute to infant death in cases where the mother is unable to properly care for her infant [15,16]. Previous research indicated that women in states that prohibit the Medicaid funding of abortions have significantly higher rates of postpartum depression than those in states that fund Medicaid abortions [17]. Mental illness among pregnant women has been associated with increased risk for infant mortality [18]. Postnatal depression has also been observed as a risk factor for sudden infant death (SIDS) [19].

Fourth, restrictive policies may encourage the delivery of infants whom the mother is unable to economically or emotionally support. For example, women from low socioeconomic backgrounds who have limited resources and income may be less able to pay to terminate their pregnancies out of pocket and more likely to carry their infant to full term [20]. Moreover, women from lower socioeconomic backgrounds may also experience limited access to prenatal care and health care, and sufficient living expenses for optimal health and growth, all of which may have an adverse effect on maternal and infant health [21]. Findings from a previous study suggests denying women abortions may be associated with greater poverty than enabling women to postpone childbearing [22].

As shown in ecological studies conducted in the US, increased state funding for family planning and abortion services is associated with lower infant mortality rates, especially for low-income women of color [7]. Conversely, restrictions on abortion services have been associated with increased infant and maternal mortality risk [23,24]. However, because these previous studies are based on aggregated data, they limit our ability to make inferences at the individual level. Moreover, few of these studies have examined interactions between restrictive abortion policy and individual-level characteristics. Therefore, the overall objective of this study is to identify the association between the number and type of state-level restrictions to abortion services during year of birth and infant mortality risk while controlling for both individual and state-level confounders. Additionally, we examine whether observed relationships are heterogeneous across socio-demographic groups, such as mother′s age, race, and educational attainment. By studying these heterogeneous associations, we aim to determine who is at greatest risk of infant mortality when such state-level abortion policies are enacted.

## 2. Methods

We obtained data from the US Cohort Linked Birth/Infant Death (LBID) Data Files on infants born 2008–2010, which is provided by the National Center for Health Statistics (NCHS). Infants were followed until their first birthdays. US state laws require birth certificates to be completed for all births. Of all deaths that occurred, around 98% were linked to the corresponding birth certificate. Instructed by US state laws, each birth requires a birth certificate to be completed, which contains information on maternal and socio-demographic characteristics and place of birth. Federal law mandates the national collection and publication of births, deaths, and other vital statistics. These data are then compiled by the National Vital Statistics System. Those with missing socio-demographic data, foreign residents, and those with records that were a mismatch between state of birth and mother′s state of residence were excluded from this analysis. Ethical approval was obtained from the University of Nevada, Reno, Institutional Review Board (code 791378-1).

## 3. Measures

### 3.1. Outcome

Infant mortality (death within 365 days of birth) was our outcome of interest.

### 3.2. Main Exposure

Our main measure of exposure was the category of restrictive state abortion law. Since the US Supreme Court′s 1973 *Roe v. Wade* decision legalizing abortion nationwide, individual states have enacted laws regulating the access and availability of abortion services in the hopes of moderating the effects of federal decision. Data on restrictive abortion laws for each US state were obtained in 2008, 2009, and 2010 from the Guttmacher Institute [25,26,27]. From 2008–2010, there were five types of state-level restrictive abortion laws directly impacting patients that the Supreme Court regarded as constitutionally permissible. These types of restrictive laws consist of: (1) Medicaid funding restrictions—prohibitions against use of state public funds to pay for abortions for indigent women [25,26,27]; (2) parental involvement laws—requirements that a parent be notified or give consent for an unmarried teen minor to obtain an abortion; (3) mandatory counselling—requirements that an abortion provider give or offer their patients information about abortion (usually written in a way to dissuade women from completing the abortion) by providing information on fetal development, fetal pain, fetal age, possible future health risks [e.g., substance abuse, breast cancer, suicide, or infertility], adoption options, and available public assistance to the birth mother); (4) mandatory waiting period—-requirements that a specified time period (usually 24 h) elapse before the procedure can be performed; (5) two-visit laws—requirements that women make two separate trips to the abortion provider prior to the procedure, elevating the monetary and time burdens for prospective patients.

Table 1 shows a complete list of the status of the five restrictive abortion laws for all 50 states and the District of Columbia in 2008. Over the next two years, several states enacted a few more abortion restrictive laws. For example, in 2008, the only restrictive law in Arizona was mandatory parental involvement [25,26,27]. By 2010, Arizona had added mandatory counselling and mandatory wait times. We looked at the relationship between each type of restriction and the number of restrictions during the year of birth of each infants and risk for infant mortality.

For this study, we included several additional state-level and individual-level covariates that could act as confounders of the relationship between restrictions to abortion services and infant mortality risk. State-level covariates include median income, proportion of population that is African-American, population size, and US census division. Individual-level maternal covariates include mother′s age, race/ethnicity, education, marital status, and nativity (US vs. foreign-born).

## 4. Statistical Analysis

Infants with missing data on their mother′s education were excluded from the analyses. We used multilevel logistic modeling (mothers and their infant nested within states) to determine the association between abortion-restriction laws during the year of birth and infant mortality risk. We tested the relationship between each of the five types of restrictive abortion laws separately and looked at the total number of restrictions (no restrictions, one or two restrictions, and three or more restrictions). The internal consistency of restrictive laws was high (Cronbach′s alpha = 0.82).

To investigate the potential effect of restrictive laws and risk of infant mortality, we adopted a step-up approach, where we systematically add variables to the models. The null model was first estimated to compute the overall predicted probability, which indicates the average probability of infant mortality across all states. Additionally, the 95% plausible value range, which is an indication of the degree of variability of the likelihood of infant mortality. For example, the plausible value range allows us to compute the range of plausible proportion (i.e., the maximum and minimum values) experiencing infant mortality across the US states. Second, we identified the crude relationship between each type of restrictive law at the year of birth (2008, 2009, or 2010) and infant mortality risk. Third, we fit our logistic regression model including state- and individual-level characteristics. Finally, we stratified the analyses by age (≤19 years, 20–25 years, 25–30 years, 30–34 years, and ≥35 years of age), race (Black vs. White), and education (less than high school vs. high school or more) to determine if relationships were heterogeneous across socio-demographic groups (results not shown).

## 5. Results

### Characteristics

Socio-demographic characteristics of infants born 2008 to 2010 are presented in Table 2. Over half of the mothers were White (53.5%), around a quarter were Hispanic (24.4%), and around 15% were Black. Just over a half of the mothers (52.5%) had some post-secondary education, almost two-thirds were married (59.0%) and roughly three-quarters were US-born (76.4%). Characteristics of the fifty states and District of Columbia also appear in Table 1. The median state-level income in 2010 was $49,973.65 (SD = 8130.60) and the average proportion of African-Americans in a state’s population was 12% (SD = 11.8).

By 2010, the most common state-level restrictive abortion law was mandatory parental involvement (*n* = 35) and the least common restrictive abortion law was the two-visit law (*n* = 7). Nine states (17.7%) had no restrictive laws, while seven (13.7%) had all five laws enacted.

From 2008 to 2010, there were 71,528 infant deaths corresponding to an infant mortality rate of 6.0 deaths/1000 births. The overall predicted probability by year was 0.61%, 0.60%, and 0.57% in 2008, 2009, and 2010, respectively. The plausible value range indicates that there is considerable variability in the cumulative incidence of infant death across US states. The plausible value range for infant mortality was 0.44–0.86%, 0.43–0.84%, and 0.40–0.80%, in 2008, 2009, and 2010, respectively.

When we tested the crude relationship between each of the five types of abortion restrictive laws, each was significantly associated with an increased odds for infant mortality (Table 3): the presence of Medicaid restrictions (OR = 1.22, 95% CI = 1.10–1.36); parental involvement (OR = 1.26, 95% CI = 1.13–1.40); mandatory counselling (OR = 1.08, 95%CI = 1.00–1.17); mandatory wait period (OR = 1.09, 95% CI = 1.01–1.18); and two-visit laws (OR = 1.18, 95% CI = 1.01–1.38). However, after adjusting for individual- and state-level characteristics, infants born in states with parental involvement laws (OR = 1.10, 95% CI = 1.02–1.19) had an increased risk for infant mortality compared to those who lived in states without such laws. No other laws remained significant when tested separately.

When we assessed the crude relationship between the total number of laws, compared to no restrictions, one or two restrictive laws (OR = 1.16, 95%CI = 1.01–1.34), and three or more restrictions (OR = 1.26, 95% CI = 1.11–1.45) were associated with an increased odds for infant mortality (Table 3). Adjusted analysis of the total number of restrictive laws resulted similar trends but resulted in decreased OR estimates. For example, infants born in states with one or two (OR = 1.08, 95%CI = 0.99–1.18) or three or more restrictive laws (OR = 1.10, 95%CI = 1.01–1.20) had greater risk for infant mortality than those born in states with no restrictive laws (Table 3).

In analyses stratified by age, in comparison to those infants born in states with no parental involvement laws, those who were born in states with this restrictive law were at greater risk for infant mortality among mothers aged ≤19 years of age (OR = 1.09, 95% CI = 1.00–1.19), and 20–25 years (OR = 1.10, 95% CI = 1.03–1.17), but not among mothers in older age categories (Table 4). No other findings were significant. When stratified by race, White infants born in states with one or two restrictions were more likely to die than those born in states with no restrictions (OR = 1.15, 95% CI = 1.04–1.27). Infants born to White mothers who were living in states with mandatory wait periods were significantly less likely to die (OR = 0.93, 95% CI = 0.88–0.99). Among Blacks, infants born in states with Medicaid Restrictions (OR = 1.08, 95% CI = 1.00–1.16) and Parental Involvement (OR = 1.08, 95% CI = 0.99–1.17) were more likely to die. Blacks were more likely to experience infant mortality in states with one or two restrictions (OR = 1.07, 95% CI = 0.97–1.19) and in states with 3 or more restrictions (OR = 1.09, 95% CI = 0.99–1.19) in comparison to those infants born in states with no restrictions. When analyses were stratified by education, infants whose mothers had less than high school education born in states with Medicaid Restrictions (OR = 1.08, 95% CI = 1.01–1.17) and one or two restrictions (OR = 1.13, 95% CI = 1.04–1.12) were more likely to die compared to those born in states with no such restrictions. Infants born to mothers with a high school education in states with Parental Involvement laws (OR = 1.11, 95% CI = 1.03–1.21) or in states with 3 or more restrictions were more likely to die (OR = 1.11, 95 %CI = 1.01–1.21) than were those born in states with no restrictions.

## 6. Discussion

Since the federal legalization of abortion in 1973, many US states have successfully enacted laws restricting access to abortion, the majority of which started in the past two decades [25,26,27]. As a result, a woman′s access to abortion services varies greatly across US states. We exploited variation in state-level restrictive abortion laws and the individual risk for infant mortality, with special attention to differential effects by maternal characteristics. We observed a significant relationship between the number of restrictive abortion laws and infant mortality risk, indicating a potential additive effect. More specifically, infants born in states with three or more restrictive laws were significantly more likely to die before their first birthday than were those born in states with no restrictions. 

We posit that the 10% increase that we observed is meaningful from a population health standpoint. There were 22,000 infant deaths in the United States in 2017, a disproportionate number of which occurred in states with restrictive abortion laws [28]. A 10% reduction in infant mortality in these states could eliminate hundreds of excess infant deaths per year.

These findings are consistent with ecological studies that have identified a significant relationship between the legalization of abortion within the US and state funding for family planning and abortion services and infant mortality rates. For example, Krieger et al. observed US infant death rates declined most quickly between 1970 and 1973 in states that legalized abortion in 1970 [29]. They also found that, since 2000, the rate ratio for infant death comparing states in the top funding quartile to states with no funding for abortion services revealed an average 15% reduction in risk among the top funders [7].

The Sexual and Reproductive Framework (SRJ) defines reproductive rights as human rights and recognizes the multiple forms of oppression that impact individuals′ decisions about their sexual and reproductive health [30]. By interpreting our results using a SRJ framework, we further emphasize the need for legislation to acknowledge how women’s sexual and reproductive health decisions are shaped by social and contextual factors as well as by individual-level resources [30]. Our results support SRJ assertions by suggesting that unintended, and differentially strong adverse effects associated with abortion policies that restrict women’s reproductive decision-making. Governments should promote sexual health policies that respect women’s reproductive decisions as part of their efforts to enhance population health [31,32]. Furthermore, it could be argued that preventing unwanted pregnancies itself and the consequent need for abortions is crucial for reproductive, maternal, and infant health. Targeting upstream factors—e.g., improved and more expansive reproductive services, especially access to contraception—could decrease the number of unwanted pregnancies, thus eliminating the need for abortions.

One reason for the observed findings could be that restrictions on abortion, through their limiting of the right to make a medical choice, could themselves have detrimental effects on the health of the mother, and consequently on the infant’s health as well. Abortion restrictions impede a woman′s ability to make health decisions and to exercise autonomy over her reproductive life. This autonomy has been identified as a fundamental human right and an important determinant of women′s health [33]. Thus, when women are compelled to carry a pregnancy to term, giving birth may have detrimental effects on the woman’s own health and, therefore, on their infant’s well-being. Additionally, social stigma attached to abortion may also have an effect on mental health of women seeking to terminate their pregnancy. For example, researchers observed that women who were denied an abortion and, therefore, carried their pregnancies to term experienced psychological distress a year later [34], which can detrimentally affect the health of their infants. Researchers may want to explore casual methods to understand whether abortion laws cause infant mortality. However, given how such laws are implemented the common assumptions of causal methods may be violated. Furthermore, future research is needed to determine whether the mechanisms linking abortion restriction laws leads to adverse infant health independently or through its effects on the mother.

In addition to directly impacting health and wellness, the state laws analyzed here may be markers for an array of factors comprising a state’s socio-political environment that fosters inequality and harms population health. In other words, states who enact restrictive abortion laws are more likely to support and fund reproductive and women′s health. This can have detrimental consequences on women′s health and therefore on, maternal and infant health [35].

We also observed significant heterogeneous associations across socio-demographic groups. When stratified by mother′s age (21 and younger and 22 and older), mandatory parental involvement, was significantly associated with an increased risk for mortality among infants of mothers from both age groups. One explanation may be that states with parental involvement laws are associated with an increase in the price of an abortion by 14% [36]. This increase in price may further act as a barrier for young mothers in obtaining an abortion. Among infants born to Black mothers, those born in states with Medicaid restrictions and parental involvement laws were more likely to die than were those born in states without these laws. Black mothers are disproportionately more likely to be from lower socioeconomic status backgrounds and are therefore less apt to have access to health services. Additionally, Black girls are more than twice as likely as White girls to become pregnant [37]. Low socioeconomic status and teenage motherhood are each risk factors for infant mortality.

One unexpected finding was that among infants born to White mothers, those in states with mandatory wait periods were significantly less likely to die in comparison to those states without this law. Mandatory wait periods may be effective at delaying an abortion, which in theory could encourage women to use unsafe practices [38]. The protective finding of mandatory wait periods against infant mortality among White mothers requires further study, including, potentially a qualitative analysis.

Infant mortality rates have been on the decline in recent years. The drop from 6.1/1000 live births in 2009 to 5.8/1000 in 2015 may be explained by decreasing birth rates among young teenage girls [39]. Infants of adolescent mothers are at greater risk for mortality than are those born to adult mothers [39]. Younger mothers may not have sufficient access to resources and health care that are essential for optimal health for their newborns. As births to teen mothers decrease, infant mortality rates could drop as a result.

### Strengths and Limitations

This study′s results should be interpreted in light of several limitations. Data analyzed for this study was collected from infants born 2008–2010, which was the most recent data available at the time of this investigation. However, one reason for using not using the most recent data is that the CDC does not release individual-level data immediately since it takes years for the data to be prepared, cleaned, and de-identified. Once this has been completed, the data is made available for public use. A delay in our investigation is further caused since obtaining state residence information requires additional paperwork and approval. Thus, linked state-individual data from 2008 to 2010, is the most temporally appropriate data to be utilized for this investigation. Since the study was not a randomized controlled trial (i.e., we could not randomize mothers into states with and without restrictions on abortion) we cannot assess whether observed associations are causal. Furthermore, because we only had data from 2008–2010 and were limited by the number of states changing or adding restrictive abortion laws, we did not utilize a quasi-experimental approach. Furthermore, endogeneity could be an issue due to residual confounding: potential confounders such as individual household income and other socioeconomic conditions were not available. Therefore, the inability to use a quasi-experimental study design and endogeneity due to residual confounding limits our ability to draw causal inferences. Lastly, although we were able to examine the moderating effects of education, we could not test whether household income acts as an effect modifier of the relationship between abortion restrictions and infant mortality.

## 7. Conclusions

Although we cannot conclude a causal relationship, results from this investigation indicate that the number of state-level restrictions on abortion may be a significant risk factor for infant mortality. In particular, mandatory parental laws that require either parental permission or notification for a minor to have an abortion are significantly associated with infant mortality. Future studies should identify the extent to which this relationship is causal. For example, researchers can take advantage of the passage of new laws against abortion, such as banning abortions once a fetal heartbeat is detected, or restricting providers with unnecessary requirements. These changes present a unique opportunity to better understand how and why these restrictions may cause adverse health outcomes for mothers and affect infant mortality risk.

## Figures and Tables

**Table 1 ijerph-17-03773-t001:** Abortion restrictions by State, 2008.

	Medicaid			Mandatory		
	Funding	Parental	Mandatory	Waiting	Two-Visit	Total
State	Restriction	Involvement	Counseling	Period	Law	Laws
Alabama	Yes	Yes	Yes	Yes	No	4
Alaska	No	No	Yes	No	No	1
Arizona	No	Yes	No	No	No	1
Arkansas	Yes	Yes	Yes	Yes	No	4
California	No	No	Yes	No	No	1
Colorado	Yes	Yes	No	No	No	2
Connecticut	No	No	Yes	No	No	1
Delaware	Yes	Yes	Yes	No	No	3
Washington DC	Yes	No	No	No	No	1
Florida	Yes	Yes	Yes	No	No	3
Georgia	Yes	Yes	Yes	Yes	No	4
Hawaii	No	No	No	No	No	0
Idaho	Yes	Yes	Yes	Yes	No	4
Illinois	No	No	No	No	No	0
Indiana	Yes	Yes	Yes	Yes	Yes	5
Iowa	Yes	Yes	No	No	No	2
Kansas	Yes	Yes	Yes	Yes	No	4
Kentucky	Yes	Yes	Yes	Yes	No	4
Louisiana	Yes	Yes	Yes	Yes	Yes	5
Maine	Yes	No	Yes	No	No	2
Maryland	No	Yes	No	No	No	1
Massachusetts	No	Yes	No	No	No	1
Michigan	Yes	Yes	Yes	Yes	No	4
Minnesota	No	Yes	Yes	Yes	No	3
Mississippi	Yes	Yes	Yes	Yes	Yes	5
Missouri	Yes	Yes	Yes	Yes	Yes	5
Montana	No	No	No	No	No	0
Nebraska	Yes	Yes	Yes	Yes	No	4
Nevada	Yes	No	Yes	No	No	2
New Hampshire	Yes	No	No	No	No	1
New Jersey	No	No	No	No	No	0
New Mexico	No	No	No	No	No	0
New York	No	No	No	No	No	0
North Carolina	Yes	Yes	No	No	No	2
North Dakota	Yes	Yes	Yes	Yes	No	4
Ohio	Yes	Yes	Yes	Yes	Yes	5
Oklahoma	Yes	Yes	Yes	Yes	No	4
Oregon	No	No	No	No	No	0
Pennsylvania	Yes	Yes	Yes	Yes	No	4
Rhode Island	Yes	Yes	Yes	No	No	3
South Carolina	Yes	Yes	Yes	Yes	No	4
South Dakota	Yes	Yes	Yes	Yes	No	4
Tennessee	Yes	Yes	Yes	No	No	3
Texas	Yes	Yes	Yes	Yes	No	4
Utah	Yes	Yes	Yes	Yes	Yes	5
Vermont	No	No	No	No	No	0
Virginia	Yes	Yes	Yes	Yes	No	4
Washington	No	No	No	No	No	0
West Virginia	No	Yes	Yes	Yes	No	3
Wisconsin	Yes	Yes	Yes	Yes	Yes	5
Wyomimg	Yes	Yes	No	No	No	2

Source. Guttmacher Institute (2008) [25].

**Table 2 ijerph-17-03773-t002:** Characteristics of mothers and US infants born 2008–2010.

Individual Level Characteristics	*n*	Percentage
Birth year		
2008	4,109,463	34.3
2009	3,993,282	33.4
2010	3,869,884	32.3
Mother′s Race		
White	6,410,568	53.5
Black	1,773,995	14.8
Native	118,496	1.0
Asian	692,386	5.8
Latin	2,922,361	24.4
Other	54,822	0.5
Education		
Less than high school	2,396,987	20.2
High School	3,277,870	27.4
Post-secondary	6,297,772	52.5
Marital Status		
Single	4,913,782	41.0
Married	7,058,847	59.0
Birth order		
First	3,986,136	33.3
Second	3,355,612	28.0
Third	2,155,132	18.0
Fourth or more	2,475,749	20.7
Mother′s nativity		
Foreign-born	2,829,786	23.6
US Born	9,142,843	76.4
US State-Level Characteristics 2010	Mean (SD)	Max, Min
Median Income, USD	49973.65 (8130.60)	(36,851, 68,854)
Population, 2010	6,054,080 (6,824,211)	(563,767, 37,300,000)
African American (%)	12 (11.8)	(0.40, 0.55)
Proportion in poverty (%)	14.8 (3.1)	(8.3, 22.4)

**Table 3 ijerph-17-03773-t003:** The relationship between abortion laws and infant mortality controlling for individual and state-level characteristics, 2008–2010.

	Odds for Infant Mortality													

	OR		OR		OR		OR		OR		OR		OR	
	95%CI		95%CI		95%CI		95%CI		95%CI		95%CI		95%CI	
**State Characteristics**														
**Abortion Laws**														
**one or two restrictions (reference group: no restrictions)**	1.08													
	(0.99,1.18)													
**3 or more restrictions (reference group: no restrictions)**	1.10													
	(1.01,1.20)													
**Medicare restrictions (reference group: no)**														
**Yes**			1.03										1.00	
			(0.96,1.11)										(0.93,1.08)	
**Parental Involvement (reference group: no)**														
**Yes**					1.10								1.11	
					(1.02,1.19)								(1.01,1.21)	
**Mandatory counselling (reference group: no)**														
**Yes**							1.04						1.04	
							(0.98,1.09)						(0.96,1.13)	
**Mandatory wait period (reference group: no)**														
**Yes**									1.03				0.97	
									(0.98,1.09)				(0.89,1.06)	
**Two visit-law (reference group: no)**														
**Yes**											1.01		0.98	
											(0.93,1.09)		(0.90,1.06)	
**State Median Income Z-score**	0.97		0.97		0.97		0.97		0.96		0.96		0.98	
	(0.93,1.02)		(0.92,1.02)		(0.93,1.02)		(0.92,1.01)		(0.92,1.01)		(0.92,1.01)		(0.93,1.03)	
**Proportion Black Z-score**	0.98		0.97		0.98		0.98		0.98		0.98		0.98	
	(0.94,1.01)		(0.94,1.01)		(0.95,1.02)		(0.94,1.02)		(0.94,1.02)		(0.94,1.02)		(0.95,1.03)	
**Proportion Poor Z-score**	0.98		0.98		0.98		0.98		0.97		0.97		0.99	
	(0.93,1.04)		(0.92,1.04)		(0.93,1.04)		(0.92,1.03)		(0.92,1.03)		(0.92,1.03)		(0.93,1.05)	
**State Population Z-score**	0.99		0.99		0.99		0.99		0.99		0.99		0.99	
	(0.97,1.01)		(0.97,1.01)		(0.97,1.01)		(0.97,1.01)		(0.97,1.01)		(0.97,1.01)		(0.97,1.01)	
**Census Division (reference group: New England)**														
**Middle Atlantic**	1.13		1.08		1.08		1.08		1.06		1.08		1.10	
	(0.98,1.29)		(0.95,1.23)		(0.95,1.21)		(0.95,1.23)		(0.93,1.21)		(0.95,1.23)		(0.96,1.25)	
**East North Central**	1.25		1.24		1.24		1.24		1.22		1.25		1.24	
	(1.11,1.41)		(1.11,1.39)		(1.08,1.42)		(1.11,1.40)		(1.08,1.38)		(1.10,1.41)		(01.09,1.40)	
**West North Central**	1.10		1.11		1.06		1.21		1.09		1.12		1.08	
	(0.98,1.23)		(0.99,1.24)		(0.95,1.19)		(1.06,1.37)		(0.97,1.23)		(1.00,1.25)		(0.95,1.21)	
**South Atlantic**	1.21		1.22		1.16		1.22		1.21		1.24		1.16	
	(1.08,1.36)		(1.09,1.37)		(1.03,1.31)		(1.09,1.37)		(1.08,1.36)		(1.10,1.39)		(1.02,1.31)	
**East South Central**	1.29		1.30		1.24		1.30		1.30		1.32		1.24	
	(1.13,1.48)		(1.14,1.49)		(1.08,1.42)		(1.14,1.49)		(1.14,1.49)		(1.16,1.51)		(1.08,1.42)	
**West South Central**	1.20		1.21		1.15		1.21		1.19		1.23		1.16	
	(1.05,1.37)		(1.06,1.38)		(1.01,1.31)		(1.06,1.37)		(1.04,1.37)		(1.08,1.39)		(1.01,1.33)	
**Mountain**	1.08		1.07		1.05		1.08		1.06		1.07		1.06	
	(0.97,1.20)		(0.96,1.19)		(0.94,1.17)		(0.96,1.20)		(0.95,1.19)		(0.96,1.20)		(0.95,1.19)	
**Pacific**	1.07		1.03		1.06		1.03		1.02		1.02		1.07	
	(0.94,1.21)		(0.91,1.16)		(0.94,1.19)		(0.91,1.16)		(0.91,1.15)		(0.91,1.15)		(0.95,1.20)	
**Individual Characteristics**														

**Birth cohort**														
**2009 (reference group: 2008)**	0.97		0.97		0.97		0.97		0.97		0.97		0.97	
	(0.95,0.99)		(0.95,0.99)		(0.95,0.99)		(0.95,0.99)		(0.95,0.99)		(0.95,0.99)		(0.95,0.99)	
**2010 (reference group: 2008)**	0.94		0.94		0.94		0.94		0.94		0.94		0.94	
	(0.93,0.96)		(0.93,0.96)		(0.93,0.96)		(0.93,0.95)		(0.93,0.96)		(0.93,0.96)		(0.93,0.96)	
**Mother′s Age (years)**														
	(0.99,1.01)		(0.99,1.01)		(0.99,1.01)		(0.99,1.01)		(0.99,1.01)		(0.99,1.01)		(0.99,1.01)	
**Mother′s Race ((reference group): white)**														
**Black**	1.88		1.88		1.88		1.88		1.88		1.88		1.88	
	(1.84,1.92)		(1.84,1.92)		(1.84,1.92)		(1.84,1.92)		(1.84,1.92)		(1.84,1.92)		(1.84,1.92)	
**Native**	1.31		1.31		1.31		1.31		1.31		1.31		1.31	
	(1.23,1.40)		(1.23,1.40)		(1.23,1.40)		(1.23,1.40)		(1.23,1.40)		(1.23,1.40)		(1.23,1.40)	
**Asian**	1.15		1.15		1.15		1.15		1.15		1.15		1.15	
	(1.10,1.20)		(1.10,1.20)		(1.10,1.20)		(1.10,1.20)		(1.10,1.20)		(1.10,1.20)		(1.10,1.20)	
**Hispanic**	0.98		0.98		0.98		0.98		0.98		0.98		0.98	
	(0.96,1.01)		(0.96,1.01)		(0.96,1.01)		(0.96,1.01)		(0.96,1.01)		(0.96,1.01)		(0.96,1.01)	
**Other**	1.25		1.25		1.25		1.25		1.25		1.25		1.25	
	(1.12,1.39)		(1.12,1.39)		(1.12,1.39)		(1.12,1.39)		(1.12,1.39)		(1.12,1.39)		(1.12,1.39)	
**Education (ref: less than high school)**														
**High School**	0.89		0.89		0.89		0.89		0.89		0.89		0.89	
	(0.87,0.91)		(0.87,0.91)		(0.87,0.91)		(0.87,0.91)		(0.87,0.91)		(0.87,0.91)		(0.87,0.91)	
**Post-Secondary**	0.68		0.68		0.68		0.68		0.68		0.68		0.68	
	(0.66,0.69)		(0.66,0.69)		(0.66,0.69)		(0.66,0.69)		(0.66,0.69)		(0.66,0.69)		(0.66,0.69)	
**With Partner (reference group: coupled)**														
**Single**	1.33		1.33		1.33		1.33		1.33		1.33		1.33	
	(1.30,1.35)		(1.32,1.35)		(1.30,1.35)		(1.30,1.35)		(1.30,1.35)		(1.30,1.35)		(1.30,1.35)	
**Nativity (reference group: born outside USA)**														
**US born**	1.30		1.30		1.30		1.31		1.30		1.30		1.30	
	(1.27,1.33)		(1.27,1.33)		(1.27,1.33)		(1.23,1.40)		(1.27,1.33)		(1.27,1.33)		(1.27,1.33)	
**Birth Order (reference group: first born)**														
**Second**	1.04		1.04		1.04		1.04		1.04		1.04		1.04	
	(1.02,1.06)		(1.02,1.06)		(1.02,1.06)		(1.02,1.06)		(1.02,1.06)		(1.02,1.06)		(1.02,1.06)	
**Third**	1.11		1.11		1.11		1.11		1.11		1.11		1.11	
	(1.08,1.14)		(1.08,1.14)		(1.08,1.14)		(1.08,1.14)		(1.08,1.14)		(1.08,1.14)		(1.08,1.14)	
	1.39		1.39		1.39		1.39		1.39		1.39		1.39	
**Fourth or more**	(1.36,1.42)		(1.36,1.42)		(1.36,1.42)		(1.36,1.42)		(1.36,1.42)		(1.36,1.42)		(1.36,1.42)	

**Table 4 ijerph-17-03773-t004:** The relationship between abortion laws and infant mortality controlling for individual and state-level characteristics, 2008–2010.

	Odds for Infant Mortality									
	≤19 Years Age		20 to 25 Years		25 to 30 Years		30 to 34 Years		35 and Older	
	OR	95%CI	OR	95%CI	OR	95%CI	OR	95%CI	OR	95%CI
**State Characteristics**										
**Parental Involvement (ref: no)**										
**Yes**	1.09	(1.00,1.19)	1.10	(1.03,1.17)	1.03	(0.94,1.12)	1.07	(0.97,1.18)	1.07	(0.98,1.16)
**State Median Income Z-score**	0.91	(0.84,0.99)	0.98	(0.92,1.04)	0.95	(0.89,1.02)	1.04	(0.95,1.14)	0.89	(0.82,0.96)
**Proportion Black Z-score**	1.05	(0.99,1.11)	0.97	(0.93,1.01)	1.00	(0.94,1.05)	0.95	(0.89,1.03)	1.05	(0.98,1.12)
**Proportion Poor Z-score**	0.88	(0.81,0.96)	0.99	(0.93,1.06)	0.99	(0.91,1.07)	0.99	(0.89,1.11)	0.87	(0.79,0.96)
**State Population Z-score**	0.99	(0.96,1.02)	0.99	(0.97,1.01)	1.00	(0.98,1.03)	0.98	(0.95,1.02)	1.02	(0.99,1.05)
**Census Division (ref: New England)**	1.00	1.00	1.00	1.00	1.00	1.00	1.00	1.00	1.00	1.00
**Middle Atlantic**	1.14	(0.96,1.35)	1.12	(0.99,1.27)	1.00	(0.87,1.15)	1.11	(0.95,1.30)	0.93	(0.82,1.07)
**East North Central**	1.17	(0.99,1.38)	1.21	(1.07,1.37)	1.18	(1.03,1.35)	1.40	(1.20,1.63)	1.15	(1.01,1.32)
**West North Central**	1.06	(0.89,1.26)	1.07	(0.93,1.22)	1.03	(0.89,1.19)	1.16	(0.98,1.37)	0.98	(0.85,1.14)
**South Atlantic**	1.06	(0.89,1.26)	1.20	(1.05,1.37)	1.11	(0.96,1.29)	1.26	(1.05,1.50)	0.97	(0.83,1.13)
**East South Central**	1.18	(0.98,1.41)	1.24	(1.06,1.44)	1.26	(1.07,1.49)	1.42	(1.18,1.70)	1.22	(1.03,1.44)
**West South Central**	1.16	(0.97,1.38)	1.14	(0.99,1.30)	1.11	(0.96,1.29)	1.38	(1.16,1.65)	1.17	(1.01,1.36)
**Mountain**	1.15	(0.97,1.36)	0.98	(0.86,1.13)	1.05	(0.91,1.21)	1.24	(1.06,1.45)	1.12	(0.97,1.29)
**Pacific**	1.14	(0.95,1.37)	1.08	(0.94,1.24)	0.99	(0.85,1.15)	1.11	(0.93,1.33)	1.02	(0.88,1.19)
**Individual Characteristics**										
**Birth cohort**										
**2008 (ref)**	1.00	1.00	1.00	1.00	1.00	1.00	1.00	1.00	1.00	1.00
**2009′**	0.95	(0.91,1.00)	0.98	(0.95,1.01)	0.98	(0.94,1.01)	0.99	(0.94,1.04)	0.94	(0.89,0.98)
**2010′**	0.91	(0.87,0.96)	0.95	(0.92,0.98)	0.96	(0.92,0.99)	1.00	(0.95,1.05)	0.89	(0.85,0.93)
**Mother′s Race (ref: white)**	1.00	1.00	1.00	1.00	1.00	1.00	1.00	1.00	1.00	1.00
**Black**	1.44	(1.37.1.51)	1.72	(1.66,1.78)	2.01	(1.93,2.10)	2.29	(2.16,2.42)	2.35	(2.22,2.49)
**Native**	0.99	(0.85,1.16)	1.34	(1.21,1.49)	1.34	(1.17,1.55)	1.39	(1.13,1.71)	1.45	(1.16,1.81)
**Asian**	1.26	(1.06,1.49)	1.07	(0.98,1.18)	1.18	(1.09,1.29)	1.26	(1.15,1.38)	1.14	(1.04,1.25)
**Hispanic**	0.83	(0.78,0.89)	0.93	(0.89,0.97)	1.04	(0.99,1.10)	1.10	(1.03,1.18)	1.13	(1.06,1.22)
**Other**	1.02	(0.76,1.36)	1.16	(0.96,1.41)	1.20	(0.96,1.51)	1.44	(1.10,1.88)	1.44	(1.13,1.84)
**Education (ref: less than high school)**	1.00	1.00	1.00	1.00	1.00	1.00	1.00	1.00	1.00	1.00
**High School**	0.86	(0.82,0.90)	0.88	(0.85,0.91)	0.93	(0.89,0.97)	1.01	(0.94,1.08)	0.98	(0.92,1.05)
**Post-Secondary**	0.82	(0.76,0.88)	0.71	(0.69,0.74)	0.73	(0.69,0.76)	0.74	(0.69,0.79)	0.68	(0.63,0.73)
**Martial Status (ref: coupled)**	1.00	1.00	1.00	1.00	1.00	1.00	1.00	1.00	1.00	1.00
**Single**	1.11	(1.04,1.19)	1.24	(1.21,1.28)	1.34	(1.30,1.39)	1.39	(1.32,1.45)	1.27	(1.21,1.33)
**Nativity (ref: born outside USA)**	1.00	1.00	1.00	1.00	1.00	1.00	1.00	1.00	1.00	1.00
**US born**	1.28	(1.18,1.38)	1.32	(1.26,1.38)	1.36	(1.30,1.43)	1.32	(1.24,1.40)	1.22	(1.15,1.29)
**Birth Order (ref: first)**	1.00	1.00	1.00	1.00	1.00	1.00	1.00	1.00	1.00	1.00
**Second**	1.49	(1.42,1.56)	1.07	(1.03,1.10)	0.96	(0.91,1.00)	0.83	(0.78,0.88)	0.79	(0.74,0.85)
**Third**	1.83	(1.70,1.98)	1.22	(1.18,1.27)	1.04	(1.00,1.10)	0.90	(0.84,0.96)	0.78	(0.73,0.84)
**Fourth or more**	1.76	(1.58,1.96)	1.54	(1.48,1.60)	1.37	(1.31,1.43)	1.12	(1.06,1.19)	0.99	(0.93,1.05)

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
