# Peer review of "Laws Restricting Access to Abortion Services and Infant Mortality Risk in the United States"

_ijerph, 2020, doi:10.3390/ijerph17113773_

Round 1

Reviewer 1 Report

a study of whether restrictive laws impact infant mortality

  • can needs to be changed to cannot - line 267
  • table 3. difficult to interpret - please revise to aid the reader
  • results seem fairly weak for such a big sample and OR CIs cross 1 but were interpreted as significant for main finding - are these results meaningful - if so justify why
  • "First, restrictive abortion policies may jeopardize patient health by limiting the range of tools that providers have to protect sick or at-risk mothers and infants. " this is stated without reference or a concrete example - i feel it would benefit from both

Author Response

Reviewer 1:

a study of whether restrictive laws impact infant mortality

  1. can needs to be changed to cannot - line 267
    • Thank you. This has been corrected
  2. table 3. difficult to interpret - please revise to aid the reader
    • We have removed the Crude Association Columns, spelled out the reference groups, and removed the 1.00 reference risk rows to declutter the table. We also corrected various typos.
  3. results seem fairly weak for such a big sample and OR CIs cross 1 but were interpreted as significant for main finding - are these results meaningful - if so justify why
    • From a population health perspective, we considered a 10% increase in infant mortality as a meaningful threshold from a policy standpoint, amounting to 100s of infant deaths per year. (See changes to discussion, 2nd paragraph). It should be noted that the models account for individual-level and state-level characteristics that may artificially blunt (overcontrol) the observed values.

The following was added to the second paragraph of the discussion:

  • We posit that the 10% increase that we observed is meaningful from a population health standpoint. There were 22,000 infant deaths in the United States in 2017, a disproportionate number of which occurred in states with restrictive abortion laws [28]. A 10% reduction in infant mortality in these states could eliminate hundreds of excess infant deaths per year.

  1. "First, restrictive abortion policies may jeopardize patient health by limiting the range of tools that providers have to protect sick or at-risk mothers and infants. " this is stated without reference or a concrete example - i feel it would benefit from both

We thank the reviewer for this comment.  We have edited this statement in the following was:

First, restrictive abortion policies may jeopardize patient health by undermining providers' medical counsel. For example, some US states require counseling that provides inaccurate information about negative mental health consequences of abortion or a link between abortion and increased risk for abortion.[10]

Reviewer 2 Report

This paper discusses the effect of laws restricting access to abortion and infant mortality risk in the U.S. Different types of abortion laws may influence infant mortality differentially, but the main conclusion is that restricting access to abortion increases the risk of infant mortality. This is not a novel conclusion but the paper makes an interesting contribution by discussing several particular laws that may influence the outcome, which could be very interesting to the readers outside U.S., unfamiliar to the legislation and how restrictive the same can be. Somehow expected, when tested separately and after controlling for confounding variables, only Parental Involvement Laws showed an effect on infant mortality. This, along with the finding when stratified by age, implies that teen mothers are at greater risk. Surprisingly, the paper does not account for the leading causes of newborn death — such as preterm birth, low birthweight, birth defects and other pregnancy complications, closely connected to access to prenatal health care but foremost to the mother’s young age, i.e., the importance of differences in health at birth. Without inclusion of these variables, it is not advisable to claim causal relationship (lines 266-267); the connection isn’t direct as abortion access can be a kind of proxy for access to all sorts of pre- and postnatal health care, not to mention correlating roughly with lower poverty rates. Another way of looking into this relationship would be looking at women with unwanted pregnancies who sought an abortion, and compare those who got one with those who did not as they would be affected by the abortion restrictions

Author Response

Reviewer 2

  1. This paper discusses the effect of laws restricting access to abortion and infant mortality risk in the U.S. Different types of abortion laws may influence infant mortality differentially, but the main conclusion is that restricting access to abortion increases the risk of infant mortality. This is not a novel conclusion but the paper makes an interesting contribution by discussing several particular laws that may influence the outcome, which could be very interesting to the readers outside U.S., unfamiliar to the legislation and how restrictive the same can be. Somehow expected, when tested separately and after controlling for confounding variables, only Parental Involvement Laws showed an effect on infant mortality. This, along with the finding when stratified by age, implies that teen mothers are at greater risk. Surprisingly, the paper does not account for the leading causes of newborn death — such as preterm birth, low birthweight, birth defects and other pregnancy complications, closely connected to access to prenatal health care but foremost to the mother’s young age, i.e., the importance of differences in health at birth. Without inclusion of these variables, it is not advisable to claim causal relationship (lines 266-267); the connection isn’t direct as abortion access can be a kind of proxy for access to all sorts of pre- and postnatal health care, not to mention correlating roughly with lower poverty rates. Another way of looking into this relationship would be looking at women with unwanted pregnancies who sought an abortion, and compare those who got one with those who did not as they would be affected by the abortion restrictions

Response:

We thank the reviewer for this suggestion. We have made the correction and we no longer make the claim that the relationship causal.

Reviewer 3 Report

The subject is interesting. There are not many articles in the literature on the subject. The contextualization of the problem is adequate. The objectives are stated correctly. The results, although very abundant, are clearly expressed. An analytical discussion is made and the limitations are reported. The main problem that the article has is the bibliography. The bibliographic references are old, most of them are over 10 years old. Also in the discussion, the social aspect of abortion should be addressed more, since the stigma that it may have for certain people (due to religious beliefs, etc.) may have an influence on the results.

Author Response

Reviewer 3

  1. The subject is interesting. There are not many articles in the literature on the subject. The contextualization of the problem is adequate. The objectives are stated correctly. The results, although very abundant, are clearly expressed. An analytical discussion is made and the limitations are reported. The main problem that the article has is the bibliography. The bibliographic references are old, most of them are over 10 years old. Also in the discussion, the social aspect of abortion should be addressed more, since the stigma that it may have for certain people (due to religious beliefs, etc.) may have an influence on the results.

Response

  • More recent references have been added to the paper.

  • We added a description of social stigma and the potential detrimental effect in the discussion (please see page 11, top):

One reason for observed findings could be that restrictions on abortion, through their limiting of the right to make a medical choice, could themselves have detrimental effects on the health of the mother, and consequently on the infant’s health as well. Abortion restrictions impede a woman's ability to make health decisions and to exercise autonomy over her reproductive life. This autonomy has been identified as a fundamental human right and an important determinant of women's health [31]. So, when women are compelled to carry a pregnancy to term, giving birth may have detrimental effects on the woman’s own health and also therefore on their infant’s well-being. Also, social stigma attached to abortion may also have an effect on mental health of women seeking to terminate their pregnancy. For example, researchers observed that women who were denied an abortion and therefore carried their pregnancies to term and experienced psychological distress year later [32], which can therefore can detrimentally affect the health of their infants.  Researchers may want to explore casual methods to understand whether abortion laws cause infant mortality.  However, given how such laws are implemented the common assumptions of causal methods may be violated. Furthermore, future research is needed to determine whether the mechanisms linking abortion restriction laws leads to adverse infant health independently or through its effects on the mother.